# Impact of surgical approaches on long-term survival outcomes of patients with pancreatic neuroendocrine carcinoma

Tingting Xia[1o], Jinhao Li[2o], Zhengqing Liu[3], Xuehua Jiao[3], Zhenguo Qiao[4*], Chunfang Xu[1*], Liangfu Han[5*]

1 Department of Gastroenterology, The First Affiliated Hospital of Soochow University, Suzhou, China, 2 Department of Pediatrics, Weifang People's Hospital, Weifang, China, 3 Department of Endocrinology, Suzhou Ninth People's Hospital, Suzhou Ninth Hospital Affiliated to Soochow University, Suzhou, China, 4 Department of Gastroenterology, Suzhou Ninth People's Hospital, Suzhou Ninth Hospital Affiliated to Soochow University, Suzhou, China, 5 Department of Emergency, The First Affiliated Hospital of Soochow University, Suzhou, China

o These authors contributed equally to this work.
* hangliangfu@suda.edu.cn (LH); qzg66666666@163.com (ZQ); xcf601@126.com (CX)

## Abstract

Surgery is the primary treatment for pancreatic neuroendocrine carcinoma (PNEC), however, the optimal surgical approach remains undetermined. We aimed to compare long-term survival outcomes between patients who received local resection (LR) and radical resection (RR) for PNEC without distant metastasis. Patients diagnosed with PNEC between 2000 and 2020 were retrieved from the Surveillance, Epidemiology, and End Results (SEER) database. Selection bias was minimized by using propensity score matching (PSM). The Kaplan-Meier method and multivariate Cox proportional hazards models were utilized to evaluate overall survival (OS) and cancer-specific survival (CSS). A total of 1331 patients were enrolled in the study, with 678 receiving LR and 653 undergoing RR. The RR group exhibited a poorer grade, larger tumor size, and TN stage compared to the LR group ($P < 0.05$). After PSM, 450 matched pairs of patients were compared, with no significant differences in demographic and clinical characteristics observed. No significant differences were observed in long-term OS ($P = 0.746$) or CSS ($P = 0.634$) between the two groups. Subgroup analyses also demonstrated comparable OS and CSS between the LR and RR groups ($P > 0.05$). Multivariate Cox analysis revealed age, AJCC stage, N stage, and chemotherapy as independent prognostic risk factors for OS, while AJCC stage and N stage were identified as independent prognostic risk factors for CSS. Our study demonstrated that in patients with PNEC without distant metastasis, LR and RR exhibit similar prognoses, suggesting that LR may be adequate as a treatment option for these patients.

## Introduction

Pancreatic Neuroendocrine Neoplasms (PNENs) constitute a rare tumor type, originating from neuroendocrine cells within the pancreas and accounting for merely 1%-2% of

**Data availability statement:** Publicly available datasets were analyzed in this study. These data can be found here: https://seer.cancer. gov/. The datasets supporting the conclusions of this article are included within the article.

**Funding:** This work was supported by the Science and Technology Development Program of Suzhou (SKYXD2022041, SYWD2024322 and SYWD2024077). The funders had no role in study design, data collection and analysis, decision to publish, or preparation of the manuscript.

**Competing interests:** The authors have declared that no competing interests exist.

all pancreatic tumors [1,2]. These neoplasms are categorized as either functional or non-functional, based on whether hormone secretion results in clinical symptoms. Functional PNENs, representing approximately 20% of cases, typically include insulinomas and gastrinomas, whereas non-functional PNENs, accounting for 75%-85% of cases, are often detected due to local compressive symptoms or during routine medical examinations [3–5]. Furthermore, PNENs can be subclassified into well-differentiated Pancreatic Neuroendocrine Tumors (PNETs) and poorly differentiated Pancreatic Neuroendocrine Carcinomas (PNEC) based on the degree of differentiation [6,7]. Surgical intervention remains the cornerstone of treatment for PNENs, particularly for localized lesions [8,9]. In cases of PNETs with concurrent liver metastases, surgical treatment is still considered if both the primary tumor and metastatic lesions can be resected simultaneously [10]. However, for patients with PNEC and distant metastases, surgical intervention has not demonstrated survival benefits, and chemotherapy is typically the preferred treatment approach [11]. For non-metastatic PNEC patients, several prominent international medical associations, including the European Neuroendocrine Tumour Society (ENETS) [12], the North American Neuroendocrine Tumour Society (NANETS) [13], and the National Comprehensive Cancer Network (NCCN) [14], advocate for surgical intervention.

The standard surgical treatments for PNENs include local tumor excision, partial pancreatic removal, pancreatoduodenectomy, and total pancreatectomy [4,11–13]. Studies have consistently shown that surgical intervention improves patient outcomes. Ye et al. [15] utilized propensity score matching (PSM) to demonstrate notably superior cancer-specific survival (CSS) in surgical patients compared to non-surgical patients with non-functional PNETs. A meta-analysis comparing 1,491 surgically managed PNETs patients with 1,607 non-surgically managed individuals revealed better overall survival (OS) in the surgical group [16]. Similarly, Fahmy et al. [16] found that surgical intervention was associated with an improved outlook for PNETs patients, and Crippa et al. [17] advocated for surgical removal in cases of both well-differentiated and poorly-differentiated PNEC that are resectable and non-metastatic. Despite the clear benefits of surgery for non-metastatic PNEC patients, there is a notable lack of published data regarding which surgical technique is more advantageous for patient prognosis. To address this knowledge gap, we conducted a study utilizing data from the Surveillance, Epidemiology, and End Results (SEER) database to compare the long-term prognosis of non-metastatic PNEC patients treated with either local resection (LR) or radical resection (RR). This study aims to provide valuable insights into the optimal surgical approach for these patients, ultimately improving their treatment outcomes.

## Materials and methods

### Patient selection

The investigation concentrated on patients who received a PNEC diagnosis between the years 2000 and 2020, leveraging data extracted from the SEER database (http://seer.cancer.gov). These patients were pinpointed using the International Classification of Diseases for Oncology, 3rd Edition (ICD-O-3), where the specific code 8246/3 denotes neuroendocrine carcinoma, NOS. The study encompassed patients who underwent surgical intervention for PNEC, but deliberately excluded those fitting particular criteria: (1) patients whose survival information remained unknown; (2) patients who did not receive surgical treatment or whose surgical procedures were unspecified; (3) patients with confirmed or unspecified distant metastases. A visual representation of the research workflow is depicted in Fig 1. The scope of the investigation embraced a multitude of factors, including sex, age, race, tumor grade, marital status, tumor size, AJCC staging, T and N stages, radiation and chemotherapy statuses, overall and

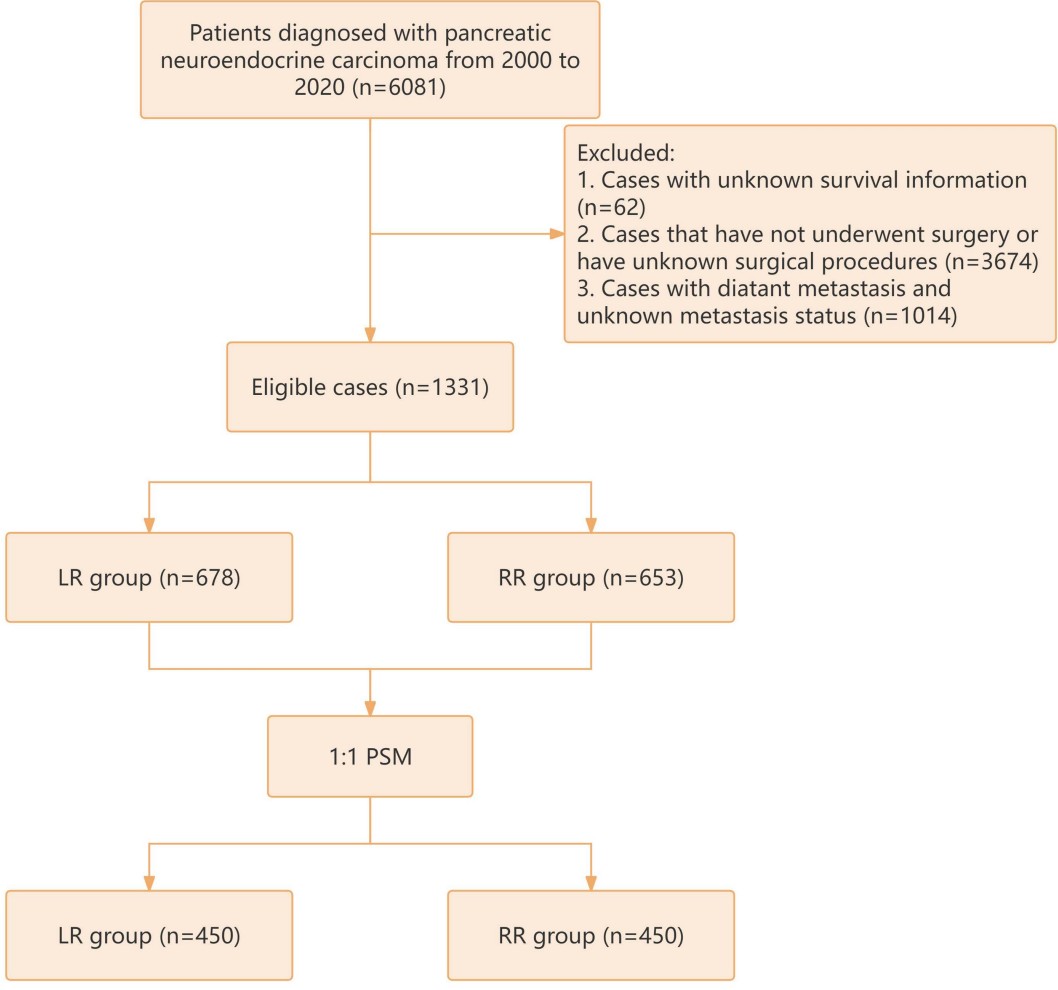

**Fig 1. Flow chart of the study.**

cancer-specific survival outcomes, as well as the duration of follow-up. According to the SEER Program's Coding and Staging Guidelines, PENC treatments were segmented into the LR and RR categories. LR entails localized tumor extraction or partial primary site removal (codes 25 and 30), whereas RR entails the complete removal of the primary site or partial/total removal of the primary site coupled with the partial/total removal of other organs (codes 35-80). The participants were stratified into two age groups: those under 60 years old (termed the younger group) and those over 60 (termed the older group). Race was classified as white, black, or other races (including American Indian, Alaska Native, and Asian/Pacific Islander). The TN stages of the cancer were delineated in accordance with the 7th edition of the AJCC Cancer Staging Manual. Marital status was dichotomized into married and unmarried, with the latter encompassing divorced, separated, single, or widowed statuses. The principal goal of the study was to gauge OS and CSS among PNEC patients. Both OS and CSS were measured from the point of PNEC diagnosis until the occurrence of death, cancer-related death, or the conclusion of follow-up, whichever event preceded. Given that the SEER database furnishes anonymized, publicly accessible data, neither Institutional Review Board (IRB) approval nor explicit consent was necessitated.

## Statistical analysis

Statistical analysis was performed using R software (version 4.1.0). Categorical variables were presented as counts and percentages, and chi-square tests were used to assess differences among groups. For continuous data that deviated from a normal distribution, the median and interquartile range (IQR) were reported, and the Mann-Whitney U test was applied for comparisons. To ensure comparability between the LR and RR groups, a 1:1 PSM approach was implemented. Variables included in the PSM model were sex, age, race, tumor grade, marital status, tumor size, AJCC stage, T and N stages, radiation status, and chemotherapy status. A caliper width of 0.01 was used to match patients. Survival analyses were conducted using the Kaplan-Meier method, and survival curves were compared with the log-rank test. To evaluate the impact of various factors on OS and CSS, Cox proportional hazards models were employed for both univariate and multivariate analyses. The choice of the Cox proportional hazards model was based on its ability to handle censored data, which is common in survival analysis, and its capability to estimate hazard ratios (HRs) for the effect of predictors on the hazard of the event (i.e., death). The proportional hazards assumption was tested using Schoenfeld residuals, and no violations were found. For the univariate analysis, each variable was individually assessed for its association with OS and CSS. Variables with a P-value less than 0.05 in the univariate analysis were considered candidates for inclusion in the multivariate model. The multivariate Cox proportional hazards model was then constructed using a stepwise backward elimination approach, retaining variables that were statistically significant at the 0.05 level. To address missing data, which is a common challenge in large databases such as the SEER database, a hybrid approach combining multiple imputation (MI) and the random forest technique was adopted. Multiple imputation was used to generate plausible values for missing data points, and the random forest algorithm was employed to impute missing values based on the observed data and the relationships among variables. This approach aimed to minimize bias and improve the accuracy of the statistical analyses. Hazard ratios (HRs) and their corresponding 95% confidence intervals (CIs) were reported for all variables included in the multivariate Cox models. Statistical significance was determined by a P-value threshold of less than 0.05.

## Results

### Patients characteristics

A comprehensive study enrolled 1331 patients, segregated into two groups: 678 in the LR category and 653 in the RR category. Before PSM, marked disparities were evident across several parameters including tumor grade, size, AJCC stage, T stage, N stage, radiation treatment status, and chemotherapy status, all statistically significant at $P < 0.05$. Notably, the RR group presented with lower tumor grades, larger tumor sizes, more advanced AJCC stages, and higher TN stages compared to the LR group. However, after conducting a 1:1 PSM, both groups were balanced with 450 patients each, and no significant differences were observed in the baseline data ($P > 0.05$) as detailed in Table 1. This table offers an exhaustive breakdown of the demographic and clinical profiles of the patients. The study also grappled with missing data issues, particularly in variables such as race (0.5% missing), grade (8.1% missing), marital status (5.3% missing), tumor size (1.8% missing), and AJCC stage (2.3% missing). To facilitate a thorough understanding, Supplementary Table 1 presents a meticulous pre-MI comparison of these variables between the two groups.

### Comparison between the LR group and RR group on OS and CSS

During the follow-up period with a median (IQR) of 76.0 (40.0, 101.0) months, 109 deaths were recorded in the LR group, with 49 of them being attributed to PNEC. In contrast, the

**Table 1. Demographic and clinical characteristics of patients before and after PSM.**

| Variables | Before PSM | | | | After PSM | | |
|---|---|---|---|---|---|---|---|
| | Total (n = 1,331) | LR group (n = 678) | RR group (n = 653) | P-value | LR group (n = 450) | RR group (n = 450) | P-value |
| **Sex, n (%)** | | | | 0.341 | | | 0.286 |
| Male | 731 (54.9%) | 381 (56.2%) | 350 (53.6%) | | 224 (49.8%) | 240 (53.3%) | |
| Female | 600 (45.1%) | 297 (43.8%) | 303 (46.4%) | | 226 (50.2%) | 210 (46.7%) | |
| **Age, years, n (%)** | | | | 0.688 | | | 0.841 |
| <60 | 623 (46.8%) | 321 (47.3%) | 302 (46.2%) | | 206 (45.8%) | 209 (46.4%) | |
| ≥60 | 708 (53.2%) | 357 (52.7%) | 351 (53.8%) | | 244 (54.2%) | 241 (53.6%) | |
| **Race, n (%)** | | | | 0.969 | | | 0.475 |
| White | 1,049 (78.8%) | 536 (79.1%) | 513 (78.6%) | | 377 (83.8%) | 363 (80.7%) | |
| Black | 144 (10.8%) | 72 (10.6%) | 72 (11.0%) | | 36 (8.0%) | 43 (9.6%) | |
| Others | 138 (10.4%) | 70 (10.3%) | 68 (10.4%) | | 37 (8.2%) | 44 (9.8%) | |
| **Grade, n (%)** | | | | **<0.001** | | | 0.949 |
| Well | 979 (73.6%) | 535 (78.9%) | 444 (68.0%) | | 361 (80.2%) | 355 (78.9%) | |
| Moderately | 244 (18.3%) | 118 (17.4%) | 126 (19.3%) | | 70 (15.6%) | 73 (16.2%) | |
| Poorly | 93 (7.0%) | 22 (3.2%) | 71 (10.9%) | | 16 (3.6%) | 19 (4.2%) | |
| Undifferentiated | 15 (1.1%) | 3 (0.4%) | 12 (1.8%) | | 3 (0.7%) | 3 (0.7%) | |
| **Marital status, n (%)** | | | | 0.513 | | | 0.720 |
| Married | 894 (67.2%) | 461 (68.0%) | 433 (66.3%) | | 305 (67.8%) | 310 (68.9%) | |
| Unmarried | 437 (32.8%) | 217 (32.0%) | 220 (33.7%) | | 145 (32.2%) | 140 (31.1%) | |
| **Tumor size, cm, n (%)** | | | | **<0.001** | | | 0.738 |
| ≤2.0 | 530 (39.8%) | 308 (45.4%) | 222 (34.0%) | | 185 (41.1%) | 183 (40.7%) | |
| 2.1–5.0 | 579 (43.5%) | 278 (41.0%) | 301 (46.1%) | | 206 (45.8%) | 200 (44.4%) | |
| >5.0 | 222 (16.7%) | 92 (13.6%) | 130 (19.9%) | | 59 (13.1%) | 67 (14.9%) | |
| **AJCC stage, n (%)** | | | | **<0.001** | | | 0.878 |
| I | 797 (59.9%) | 470 (69.3%) | 327 (50.1%) | | 302 (67.1%) | 307 (68.2%) | |
| II | 491 (36.9%) | 196 (28.9%) | 295 (45.2%) | | 143 (31.8%) | 137 (30.4%) | |
| III | 43 (3.2%) | 12 (1.8%) | 31 (4.7%) | | 5 (1.1%) | 6 (1.3%) | |
| **T stage, n (%)** | | | | **<0.001** | | | 0.925 |
| T1 | 479 (36.0%) | 285 (42.0%) | 194 (29.7%) | | 175 (38.9%) | 170 (37.8%) | |
| T2 | 452 (34.0%) | 242 (35.7%) | 210 (32.2%) | | 172 (38.2%) | 177 (39.3%) | |
| T3 | 354 (26.6%) | 132 (19.5%) | 222 (34.0%) | | 98 (21.8%) | 97 (21.6%) | |
| T4 | 22 (1.7%) | 3 (0.4%) | 19 (2.9%) | | 2 (0.4%) | 1 (0.2%) | |
| TX | 24 (1.8%) | 16 (2.4%) | 8 (1.2%) | | 3 (0.7%) | 5 (1.1%) | |
| **N stage, n (%)** | | | | **<0.001** | | | 0.818 |
| N0 | 994 (74.7%) | 558 (82.3%) | 436 (66.8%) | | 360 (80.0%) | 360 (80.0%) | |
| N1 | 320 (24.0%) | 110 (16.2%) | 210 (32.2%) | | 85 (18.9%) | 87 (19.3%) | |
| NX | 17 (1.3%) | 10 (1.5%) | 7 (1.1%) | | 5 (1.1%) | 3 (0.7%) | |
| **Radiation, n (%)** | | | | **0.005** | | | 1.000 |
| None/Unknown | 1,300 (97.7%) | 670 (98.8%) | 630 (96.5%) | | 444 (98.7%) | 444 (98.7%) | |
| Yes | 31 (2.3%) | 8 (1.2%) | 23 (3.5%) | | 6 (1.3%) | 6 (1.3%) | |
| **Chemotherapy, n (%)** | | | | **<0.001** | | | 0.458 |
| No/Unknown | 1,248 (93.8%) | 657 (96.9%) | 591 (90.5%) | | 437 (97.1%) | 433 (96.2%) | |
| Yes | 83 (6.2%) | 21 (3.1%) | 62 (9.5%) | | 13 (2.9%) | 17 (3.8%) | |
| **Survival months, median (IQR)** | 77.0 (34.0, 102.0) | 76.0 (40.0, 101.0) | 78.0 (30.0, 102.0) | 0.315 | 78.0 (40.0, 103.0) | 80.0 (34.0, 103.0) | 0.511 |

LR: local resection; RR: radical resection; PSM: propensity score matching; Others: American Indian, Alaska Native, Asian/Pacifc Islander; IQR: interquartile range; bold values indicate P < 0.05.

RR group had 141 deaths, 82 of which were due to PNEC. Before PSM, the RR group showed significantly worse OS (HR 1.38, 95% CI 1.08-1.78, *P* = 0.011) (Fig 2A) and CSS (HR 1.79, 95% CI 1.26-2.55, *P* = 0.001) (Fig 2B) than the LR group. However, after PSM, the OS (HR 1.06, 95% CI 0.76-1.46, *P* = 0.746) (Fig 2C) and CSS (HR 1.12, 95% CI 0.70-1.78, *P* = 0.634) (Fig 2D) of the RR group were comparable to those of the LR group.

## Univariate and multivariate cox regression

Prior to PSM, a univariate Cox regression analysis identified several factors including age, race, tumor grade, AJCC stage, T stage, N stage, surgical modality, radiation status, and chemotherapy status as having an independent influence on OS among PNEC patients. This was further supported by a multivariate Cox regression analysis, which pinpointed age, race, tumor grade, AJCC stage, and chemotherapy status as key independent predictors of OS in this patient cohort (Table 2). At the same time, a univariate Cox regression analysis found that age, tumor grade, tumor size, AJCC stage, T stage, N stage, surgical modality, radiation status, and chemotherapy status each had an independent impact on CSS in PNEC patients. A subsequent multivariate Cox regression analysis reinforced the importance of age, tumor grade, AJCC stage, radiation status, and chemotherapy status as crucial independent predictors of CSS in this patient population (Table 3).

After PSM, a univariate Cox regression analysis revealed that factors such as age, tumor grade, AJCC stage, T stage, N stage, radiation status, and chemotherapy status, excluding surgical modality, had an independent effect on OS in PNEC patients. A multivariate Cox regression analysis further validated that age, AJCC stage, N stage, and chemotherapy were key independent predictors of OS in this specific patient group (Table 4). Likewise, a univariate Cox regression analysis indicated that tumor grade, tumor size, AJCC stage, T stage, N stage, radiation status, and chemotherapy status, excluding surgical modality, each had an independent influence on CSS in PNEC patients. A multivariate Cox regression analysis subsequently underscored the significance of AJCC stage and N stage as crucial independent predictors of CSS in this particular patient cohort (Table 5).

## Subgroup analysis

After PSM, both the LR group and the RR group exhibited comparable OS and CSS (*P* > 0.05). To further explore the consistency of these findings across different patient characteristics, we conducted subgroup analyses based on sex, age, race, grade, marital status, tumor size, AJCC stage, T stage, N stage, radiation, and chemotherapy. In these subgroup analyses, we observed that the OS and CSS remained comparable between the LR and RR groups across all stratification factors (*P* > 0.05). It is important to note that while some subgroups had relatively smaller sample sizes (e.g., certain age groups, specific tumor sizes), the statistical non-significance consistently observed across all subgroups suggests that the primary finding of similar prognoses between LR and RR is robust. The clinical importance of these findings lies in their implication for treatment decision-making. Our study suggests that for patients with PNEC without distant metastasis, the choice between LR and RR may not significantly impact long-term survival outcomes. This is particularly relevant in clinical scenarios where preserving organ function and minimizing surgical morbidity are crucial considerations. Fig 3 and Fig 4 provide detailed information on the comparisons among different subgroups, illustrating the consistency of our findings across various patient characteristics. These figures visually represent the lack of significant differences in OS and CSS between the LR and RR groups across all subgroups analyzed.

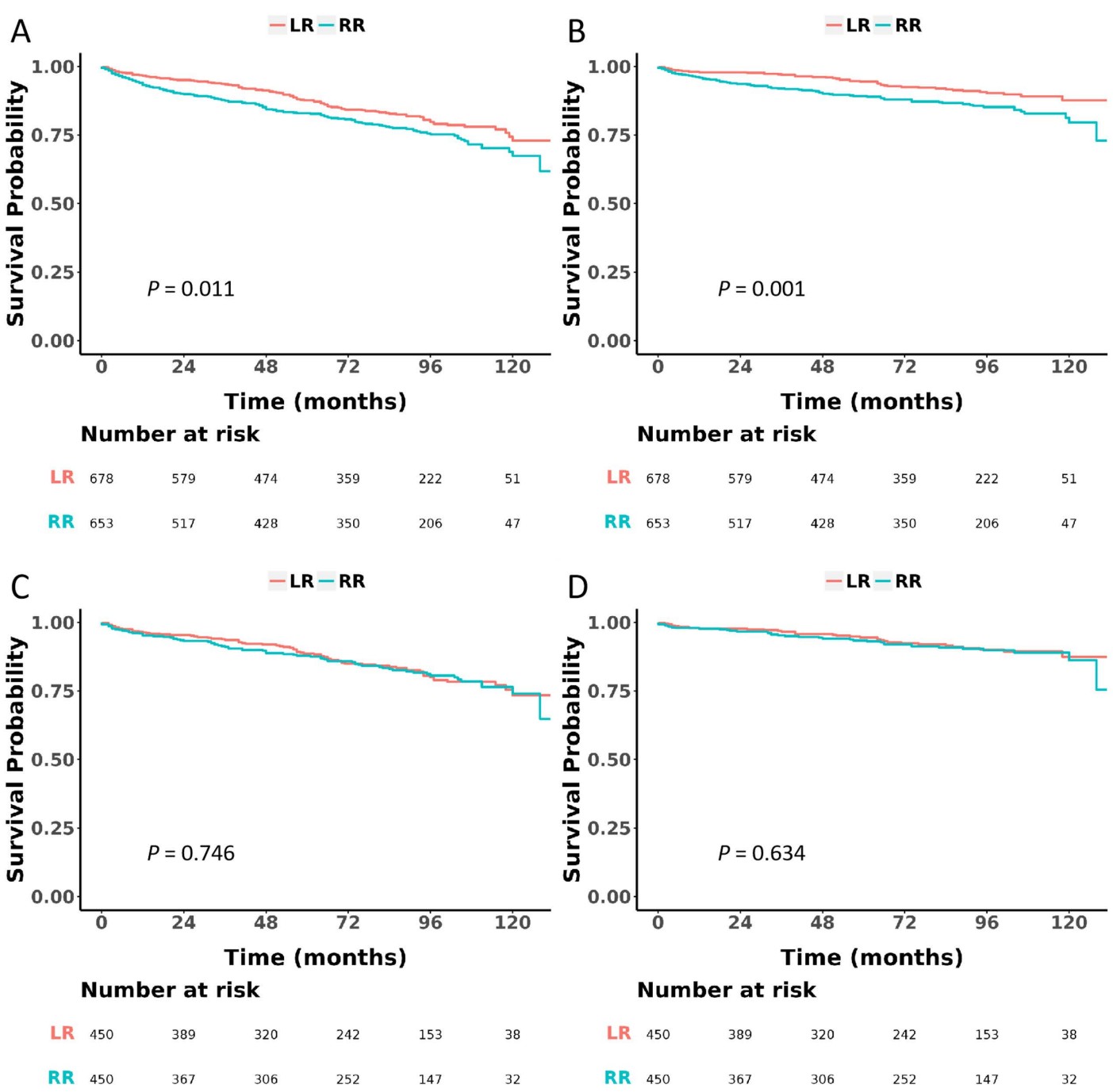

**Fig 2. Overall survival (OS) and cancer-specific survival (CSS) were compared between local resection (LR) group and radical resection (RR) group before and after propensity score matching (PSM)** A. OS before PSM; B CSS before PSM; C OS after PSM; D CSS after PSM.

## Discussion

In this study, we leveraged the SEER database to investigate the most effective surgical approach for PNEC without distant metastasis by contrasting the outcomes of LR versus RR.

**Table 2. Univariate and multivariate cox regression for analyzing the overall survival for patients with PNEC before PSM.**

| | Univariate | | | Multivariate | | |
|---|---|---|---|---|---|---|
| | HR | 95%CI | *P*-value | HR | 95%CI | *P*-value |
| **Sex** | | | | | | |
| Male | 1 | Reference | | – | – | – |
| Female | 0.82 | 0.64, 1.06 | 0.127 | – | – | – |
| **Age, years** | | | | | | |
| <60 | 1 | Reference | | 1 | Reference | |
| ≥60 | 2.45 | 1.86, 3.23 | **<0.001** | 2.65 | 2.00, 3.52 | **<0.001** |
| **Race** | | | | | | |
| White | 1 | Reference | | 1 | Reference | |
| Black | 0.88 | 0.57, 1.34 | 0.543 | 1.18 | 0.77, 1.83 | 0.442 |
| Others | 0.53 | 0.32, 0.88 | **0.015** | 0.52 | 0.31, 0.87 | **0.013** |
| **Grade** | | | | | | |
| Well | 1 | Reference | | 1 | Reference | |
| Moderately | 1.06 | 0.75, 1.50 | 0.748 | 0.90 | 0.63, 1.29 | 0.568 |
| Poorly | 3.94 | 2.78, 5.58 | **<0.001** | 1.87 | 1.17, 3.00 | **0.009** |
| Undifferentiated | 5.21 | 2.56, 10.62 | **<0.001** | 4.71 | 2.24, 9.90 | **<0.001** |
| **Marital status** | | | | | | |
| Married | 1 | Reference | | – | – | – |
| Unmarried | 1.26 | 0.97, 1.63 | 0.085 | – | – | – |
| **Tumor size, cm** | | | | | | |
| ≤2.0 | 1 | Reference | | 1 | Reference | |
| 2.1–5.0 | 1.25 | 0.94, 1.65 | 0.129 | 1.21 | 0.66, 2.22 | 0.538 |
| >5.0 | 1.56 | 1.11, 2.19 | 0.011 | 1.09 | 0.57, 2.07 | 0.792 |
| **AJCC stage** | | | | | | |
| I | 1 | Reference | | 1 | Reference | |
| II | 2.14 | 1.65, 2.77 | **<0.001** | 2.50 | 1.42, 4.41 | **0.002** |
| III | 2.81 | 1.61, 4.92 | **<0.001** | 1.77 | 1.18, 3.26 | **0.013** |
| **T stage** | | | | | | |
| T1 | 1 | Reference | | 1 | Reference | |
| T2 | 0.96 | 0.69, 1.35 | 0.827 | 0.78 | 0.39, 1.56 | 0.484 |
| T3 | 1.95 | 1.43, 2.65 | **<0.001** | 0.74 | 0.35, 1.56 | 0.431 |
| T4 | 2.79 | 1.39, 5.58 | **0.004** | 2.40 | 0.48, 11.87 | 0.285 |
| TX | 3.20 | 1.60, 6.41 | **0.001** | 1.79 | 0.55, 5.79 | 0.332 |
| **N stage** | | | | | | |
| N0 | 1 | Reference | | 1 | Reference | |
| N1 | 1.64 | 1.26, 2.14 | **<0.001** | 0.77 | 0.52, 1.14 | 0.188 |
| NX | 4.64 | 2.28, 9.45 | **<0.001** | 1.25 | 0.47, 3.36 | 0.656 |
| **Surgical modality** | | | | | | |
| LR | 1 | Reference | | 1 | Reference | |
| RR | 1.38 | 1.08, 1.78 | **0.011** | 1.07 | 0.82, 1.40 | 0.599 |
| **Radiation** | | | | | | |
| Yes | 1 | Reference | | 1 | Reference | |
| No/Unknown | 2.83 | 1.68, 4.77 | **<0.001** | 1.33 | 0.75, 2.36 | 0.330 |
| **Chemotherapy** | | | | | | |
| Yes | 1 | Reference | | 1 | Reference | |
| No/Unknown | 4.01 | 2.90, 5.57 | **<0.001** | 2.09 | 1.35, 3.24 | **<0.001** |

PNEC: pancreatic neuroendocrine carcinoma; LR: local resection; RR: radical resection; PSM: propensity score matching; Others: American Indian, Alaska Native, Asian/Pacifc Islander; HR: hazard ratios; bold values indicate *P* < 0.05.

**Table 3. Univariate and multivariate cox regression for analyzing the cancer-specific survival for patients with PNEC before PSM.**

| | Univariate | | | Multivariate | | |
|---|---|---|---|---|---|---|
| | HR | 95%CI | *P*-value | HR | 95%CI | *P*-value |
| **Sex** | | | | | | |
| Male | 1 | Reference | | – | – | – |
| Female | 0.83 | 0.58, 1.17 | 0.283 | – | – | – |
| **Age, years** | | | | | | |
| <60 | 1 | Reference | | 1 | Reference | |
| ≥60 | 1.62 | 1.13, 2.30 | **0.008** | 1.83 | 1.28, 2.64 | **0.001** |
| **Race** | | | | | | |
| White | 1 | Reference | | – | – | – |
| Black | 0.84 | 0.46, 1.53 | 0.571 | – | – | – |
| Others | 0.65 | 0.34, 1.24 | 0.187 | – | – | – |
| **Grade** | | | | | | |
| Well | 1 | Reference | | 1 | Reference | |
| Moderately | 1.27 | 0.79, 2.05 | 0.318 | 0.95 | 0.58, 1.55 | 0.842 |
| Poorly | 6.53 | 4.28, 9.98 | **<0.001** | 2.03 | 1.14, 3.63 | **0.017** |
| Undifferentiated | 5.58 | 2.04, 15.31 | **<0.001** | 3.75 | 1.31, 10.77 | **0.014** |
| **Marital status** | | | | | | |
| Married | 1 | Reference | | – | – | – |
| Unmarried | 1.41 | 0.99, 2.00 | 0.055 | – | – | – |
| **Tumor size, cm** | | | | | | |
| ≤2.0 | 1 | Reference | | 1 | Reference | |
| 2.1–5.0 | 2.19 | 1.42, 3.39 | **<0.001** | 1.25 | 0.57, 2.72 | 0.576 |
| >5.0 | 2.67 | 1.62, 4.40 | **<0.001** | 1.05 | 0.46, 2.39 | 0.915 |
| **AJCC stage** | | | | | | |
| I | 1 | Reference | | 1 | Reference | |
| II | 4.58 | 3.07, 6.82 | **<0.001** | 4.33 | 2.06, 9.08 | **<0.001** |
| III | 5.01 | 2.31, 10.85 | **<0.001** | 3.01 | 1.13, 8.06 | **0.003** |
| **T stage** | | | | | | |
| T1 | 1 | Reference | | 1 | Reference | |
| T2 | 1.59 | 0.92, 2.74 | 0.099 | 1.08 | 0.42, 2.79 | 0.881 |
| T3 | 4.40 | 2.70, 7.19 | **<0.001** | 0.95 | 0.35, 2.57 | 0.925 |
| T4 | 6.11 | 2.47, 15.15 | **<0.001** | 3.83 | 0.40, 36.99 | 0.246 |
| TX | 4.56 | 1.56, 13.30 | **0.005** | 1.62 | 0.29, 9.10 | 0.585 |
| **N stage** | | | | | | |
| N0 | 1 | Reference | | 1 | Reference | |
| N1 | 2.47 | 1.74, 3.50 | **<0.001** | 0.75 | 0.47, 1.21 | 0.236 |
| NX | 5.02 | 1.83, 13.75 | **0.002** | 1.54 | 0.38, 6.29 | 0.550 |
| **Surgical modality** | | | | | | |
| LR | 1 | Reference | | 1 | Reference | |
| RR | 1.79 | 1.26, 2.55 | **0.001** | 1.10 | 0.75, 1.60 | 0.635 |
| **Radiation** | | | | | | |
| Yes | 1 | Reference | | 1 | Reference | |
| No/Unknown | 5.73 | 3.34, 9.81 | **<0.001** | 1.93 | 1.05, 3.54 | **0.034** |
| **Chemotherapy** | | | | | | |
| Yes | 1 | Reference | | 1 | Reference | |
| No/Unknown | 6.76 | 4.58, 9.97 | **<0.001** | 2.54 | 1.49, 4.32 | **<0.001** |

PNEC: pancreatic neuroendocrine carcinoma; LR: local resection; RR: radical resection; PSM: propensity score matching; Others: American Indian, Alaska Native, Asian/Pacifc Islander; HR: hazard ratios; bold values indicate *P* < 0.05.

**Table 4. Univariate and multivariate cox regression for analyzing the overall survival for patients with PNEC after PSM.**

| | Univariate | | | Multivariate | | |
|---|---|---|---|---|---|---|
| | HR | 95%CI | *P*-value | HR | 95%CI | *P*-value |
| **Sex** | | | | | | |
| Male | 1 | Reference | | – | – | – |
| Female | 0.77 | 0.56, 1.07 | 0.121 | – | – | – |
| **Age, years** | | | | | | |
| <60 | 1 | Reference | | 1 | Reference | |
| ≥60 | 1.99 | 1.40, 2.83 | **<0.001** | 2.17 | 1.50, 3.14 | **<0.001** |
| **Race** | | | | | | |
| White | 1 | Reference | | – | – | – |
| Black | 0.74 | 0.39, 1.41 | 0.366 | – | – | – |
| Others | 0.47 | 0.22, 1.00 | 0.050 | – | – | – |
| **Grade** | | | | | | |
| Well | 1 | Reference | | 1 | Reference | |
| Moderately | 0.72 | 0.42, 1.21 | 0.214 | 0.63 | 0.36, 1.08 | 0.095 |
| Poorly | 4.34 | 2.53, 7.46 | **<0.001** | 2.16 | 0.92, 5.06 | 0.077 |
| Undifferentiated | 3.64 | 0.90, 14.82 | 0.071 | 3.19 | 0.73, 13.87 | 0.122 |
| **Marital status** | | | | | | |
| Married | 1 | Reference | | – | – | – |
| Unmarried | 1.19 | 0.85, 1.68 | 0.311 | – | – | – |
| **Tumor size, cm** | | | | | | |
| ≤2.0 | 1 | Reference | | – | – | – |
| 2.1–5.0 | 1.06 | 0.74, 1.52 | 0.764 | – | – | – |
| >5.0 | 1.32 | 0.83, 2.09 | 0.241 | – | – | – |
| **AJCC stage** | | | | | | |
| I | 1 | Reference | | 1 | Reference | |
| II | 2.03 | 1.46, 2.83 | **<0.001** | 3.82 | 1.71, 8.54 | **0.001** |
| III | 4.96 | 2.28, 10.78 | **<0.001** | 2.16 | 1.49, 9.54 | **0.009** |
| **T stage** | | | | | | |
| T1 | 1 | Reference | | 1 | Reference | |
| T2 | 0.78 | 0.52, 1.19 | 0.251 | 0.83 | 0.54, 1.28 | 0.400 |
| T3 | 1.88 | 1.26, 2.79 | **0.002** | 0.69 | 0.33, 1.47 | 0.342 |
| T4 | 8.64 | 2.68, 27.81 | **<0.001** | 2.46 | 0.40, 15.14 | 0.331 |
| TX | 2.92 | 1.05, 8.13 | **0.040** | | | |
| **N stage** | | | | | | |
| N0 | 1 | Reference | | 1 | Reference | |
| N1 | 1.22 | 0.83, 1.80 | 0.313 | 1.48 | 1.27, 1.87 | **0.015** |
| NX | 4.23 | 1.72, 10.37 | **0.002** | 0.78 | 0.20, 3.07 | 0.724 |
| **Surgical modality** | | | | | | |
| LR | 1 | Reference | | – | – | – |
| RR | 1.06 | 0.76, 1.46 | 0.746 | – | – | – |
| **Radiation** | | | | | | |
| Yes | 1 | Reference | | 1 | Reference | |
| No/Unknown | 3.58 | 1.47, 8.76 | **0.005** | 1.71 | 0.65, 4.49 | 0.275 |
| **Chemotherapy** | | | | | | |
| Yes | 1 | Reference | | 1 | Reference | |
| No/Unknown | 5.10 | 3.07, 8.49 | **<0.001** | 2.32 | 1.02, 5.29 | **0.046** |

PNEC: pancreatic neuroendocrine carcinoma; LR: local resection; RR: radical resection; PSM: propensity score matching; Others: American Indian, Alaska Native, Asian/Pacifc Islander; HR: hazard ratios; bold values indicate *P* < 0.05.

**Table 5. Univariate and multivariate cox regression for analyzing the cancer-specific survival for patients with PNEC after PSM.**

| | Univariate | | | Multivariate | | |
|---|---|---|---|---|---|---|
| | HR | 95%CI | *P*-value | HR | 95%CI | *P*-value |
| **Sex** | | | | – | – | – |
| Male | 1 | Reference | | – | – | – |
| Female | 0.64 | 0.40, 1.04 | 0.071 | – | – | – |
| **Age, years** | | | | | | |
| <60 | 1 | Reference | | – | – | – |
| ≥60 | 1.28 | 0.80, 2.05 | 0.312 | – | – | – |
| **Race** | | | | | | |
| White | 1 | Reference | | – | – | – |
| Black | 0.61 | 0.22, 1.67 | 0.333 | – | – | – |
| Others | 0.55 | 0.20, 1.52 | 0.250 | – | – | – |
| **Grade** | | | | | | |
| Well | 1 | Reference | | 1 | Reference | |
| Moderately | 0.92 | 0.45, 1.88 | 0.825 | 0.67 | 0.32, 1.41 | 0.291 |
| Poorly | 7.77 | 4.11, 14.69 | **<0.001** | 3.12 | 0.83, 9.43 | 0.144 |
| Undifferentiated | 4.03 | 0.55, 29.38 | 0.169 | 2.76 | 0.32, 23.96 | 0.356 |
| **Marital status** | | | | | | |
| Married | 1 | Reference | | – | – | – |
| Unmarried | 1.52 | 0.94, 2.45 | 0.084 | – | – | – |
| **Tumor size, cm** | | | | | | |
| ≤2.0 | 1 | Reference | | 1 | Reference | |
| 2.1–5.0 | 1.72 | 0.99, 2.99 | 0.057 | 0.82 | 0.32, 2.14 | 0.687 |
| >5.0 | 2.26 | 1.16, 4.39 | **0.016** | 0.78 | 0.28, 2.18 | 0.633 |
| **AJCC stage** | | | | | | |
| I | 1 | Reference | | 1 | Reference | |
| II | 4.80 | 2.86, 8.04 | **<0.001** | 8.26 | 2.90, 23.56 | **<0.001** |
| III | 9.66 | 3.31, 28.23 | **<0.001** | 4.31 | 1.59, 31.45 | **0.029** |
| **T stage** | | | | | | |
| T1 | 1 | Reference | | 1 | Reference | |
| T2 | 1.34 | 0.66, 2.71 | 0.417 | 1.56 | 0.47, 5.18 | 0.467 |
| T3 | 4.87 | 2.58, 9.20 | **<0.001** | 1.13 | 0.31, 4.15 | 0.855 |
| T4 | 19.99 | 4.50, 88.85 | **<0.001** | 1.70 | 0.16, 18.00 | 0.660 |
| TX | 5.39 | 1.21, 24.02 | **0.027** | | | |
| **N stage** | | | | | | |
| N0 | 1 | Reference | | 1 | Reference | |
| N1 | 1.92 | 1.16, 3.17 | **0.011** | 1.35 | 1.16, 1.76 | **0.008** |
| NX | 3.98 | 0.97, 16.41 | 0.056 | 0.75 | 0.11, 5.19 | 0.770 |
| **Surgical modality** | | | | | | |
| LR | 1 | Reference | | – | – | – |
| RR | 1.12 | 0.70, 1.78 | 0.634 | – | – | – |
| **Radiation** | | | | | | |
| Yes | 1 | Reference | | 1 | Reference | |
| No/Unknown | 7.48 | 3.00, 18.65 | **<0.001** | 2.38 | 0.85, 6.64 | 0.099 |
| **Chemotherapy** | | | | | | |
| Yes | 1 | Reference | | 1 | Reference | |
| No/Unknown | 8.43 | 4.60, 15.47 | **<0.001** | 2.76 | 0.91, 8.39 | 0.073 |

PNEC: pancreatic neuroendocrine carcinoma; LR: local resection; RR: radical resection; PSM: propensity score matching; Others: American Indian, Alaska Native, Asian/Pacifc Islander; HR: hazard ratios; bold values indicate *P* < 0.05.

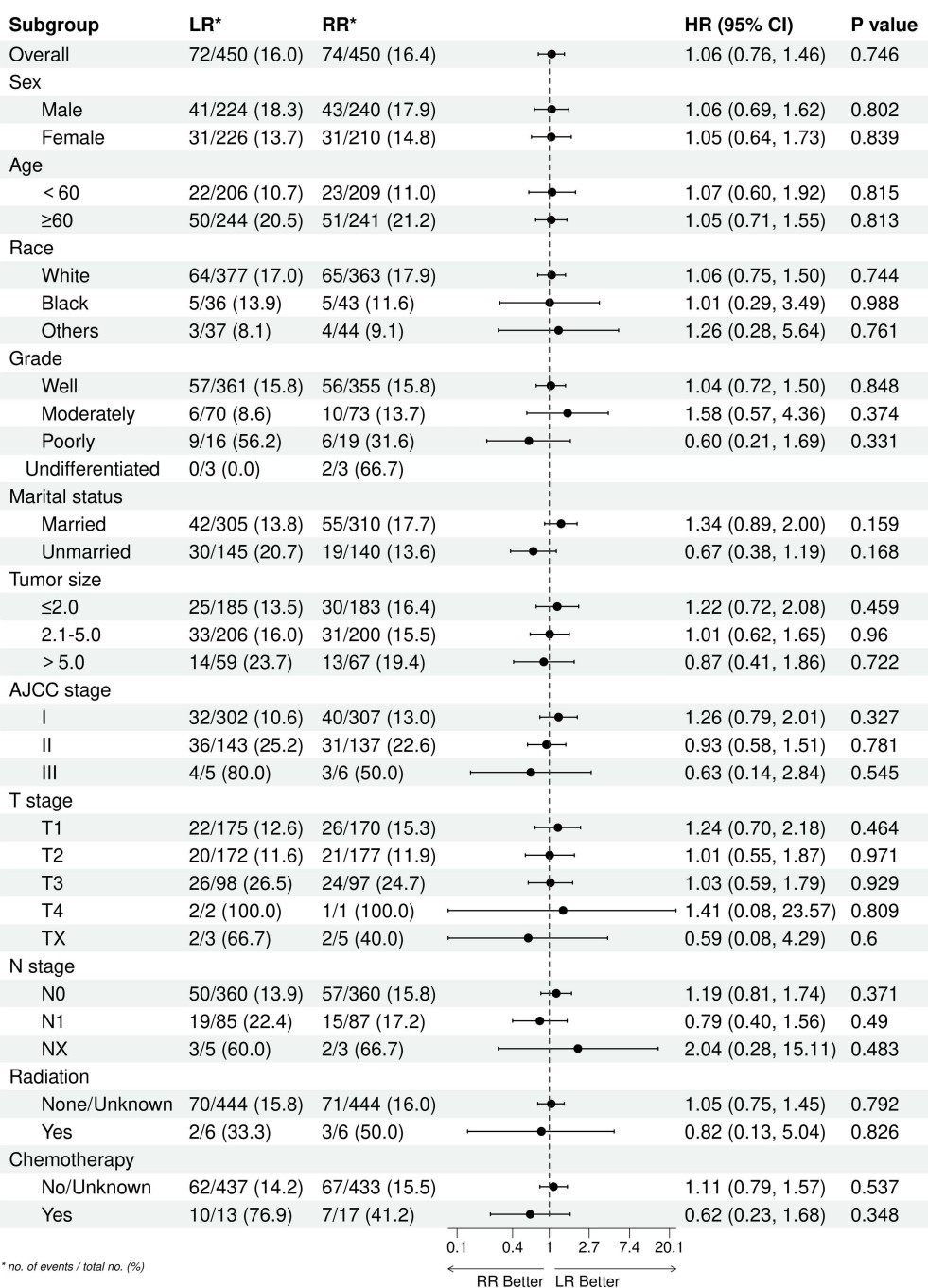

| Subgroup | LR* | RR* | | HR (95% CI) | P value |
|---|---|---|---|---|---|
| Overall | 72/450 (16.0) | 74/450 (16.4) | | 1.06 (0.76, 1.46) | 0.746 |
| Sex | | | | | |
| Male | 41/224 (18.3) | 43/240 (17.9) | | 1.06 (0.69, 1.62) | 0.802 |
| Female | 31/226 (13.7) | 31/210 (14.8) | | 1.05 (0.64, 1.73) | 0.839 |
| Age | | | | | |
| < 60 | 22/206 (10.7) | 23/209 (11.0) | | 1.07 (0.60, 1.92) | 0.815 |
| ≥60 | 50/244 (20.5) | 51/241 (21.2) | | 1.05 (0.71, 1.55) | 0.813 |
| Race | | | | | |
| White | 64/377 (17.0) | 65/363 (17.9) | | 1.06 (0.75, 1.50) | 0.744 |
| Black | 5/36 (13.9) | 5/43 (11.6) | | 1.01 (0.29, 3.49) | 0.988 |
| Others | 3/37 (8.1) | 4/44 (9.1) | | 1.26 (0.28, 5.64) | 0.761 |
| Grade | | | | | |
| Well | 57/361 (15.8) | 56/355 (15.8) | | 1.04 (0.72, 1.50) | 0.848 |
| Moderately | 6/70 (8.6) | 10/73 (13.7) | | 1.58 (0.57, 4.36) | 0.374 |
| Poorly | 9/16 (56.2) | 6/19 (31.6) | | 0.60 (0.21, 1.69) | 0.331 |
| Undifferentiated | 0/3 (0.0) | 2/3 (66.7) | | | |
| Marital status | | | | | |
| Married | 42/305 (13.8) | 55/310 (17.7) | | 1.34 (0.89, 2.00) | 0.159 |
| Unmarried | 30/145 (20.7) | 19/140 (13.6) | | 0.67 (0.38, 1.19) | 0.168 |
| Tumor size | | | | | |
| ≤2.0 | 25/185 (13.5) | 30/183 (16.4) | | 1.22 (0.72, 2.08) | 0.459 |
| 2.1-5.0 | 33/206 (16.0) | 31/200 (15.5) | | 1.01 (0.62, 1.65) | 0.96 |
| > 5.0 | 14/59 (23.7) | 13/67 (19.4) | | 0.87 (0.41, 1.86) | 0.722 |
| AJCC stage | | | | | |
| I | 32/302 (10.6) | 40/307 (13.0) | | 1.26 (0.79, 2.01) | 0.327 |
| II | 36/143 (25.2) | 31/137 (22.6) | | 0.93 (0.58, 1.51) | 0.781 |
| III | 4/5 (80.0) | 3/6 (50.0) | | 0.63 (0.14, 2.84) | 0.545 |
| T stage | | | | | |
| T1 | 22/175 (12.6) | 26/170 (15.3) | | 1.24 (0.70, 2.18) | 0.464 |
| T2 | 20/172 (11.6) | 21/177 (11.9) | | 1.01 (0.55, 1.87) | 0.971 |
| T3 | 26/98 (26.5) | 24/97 (24.7) | | 1.03 (0.59, 1.79) | 0.929 |
| T4 | 2/2 (100.0) | 1/1 (100.0) | | 1.41 (0.08, 23.57) | 0.809 |
| TX | 2/3 (66.7) | 2/5 (40.0) | | 0.59 (0.08, 4.29) | 0.6 |
| N stage | | | | | |
| N0 | 50/360 (13.9) | 57/360 (15.8) | | 1.19 (0.81, 1.74) | 0.371 |
| N1 | 19/85 (22.4) | 15/87 (17.2) | | 0.79 (0.40, 1.56) | 0.49 |
| NX | 3/5 (60.0) | 2/3 (66.7) | | 2.04 (0.28, 15.11) | 0.483 |
| Radiation | | | | | |
| None/Unknown | 70/444 (15.8) | 71/444 (16.0) | | 1.05 (0.75, 1.45) | 0.792 |
| Yes | 2/6 (33.3) | 3/6 (50.0) | | 0.82 (0.13, 5.04) | 0.826 |
| Chemotherapy | | | | | |
| No/Unknown | 62/437 (14.2) | 67/433 (15.5) | | 1.11 (0.79, 1.57) | 0.537 |
| Yes | 10/13 (76.9) | 7/17 (41.2) | | 0.62 (0.23, 1.68) | 0.348 |

0.1    0.4    1    2.7    7.4   20.1

← RR Better    LR Better →

*no. of events / total no. (%)

**Fig 3. Subgroup analysis of OS between LR group and RR group after PSM.**

Our results notably revealed no survival benefit for RR over LR in patients with PNEC. Prior to PSM analysis, the RR group showed more advanced disease characteristics compared to the LR group (*P* < 0.05). Nevertheless, after PSM, no significant differences were detected in long-term OS or CSS between the two groups. Additionally, multivariate Cox analysis pinpointed age, AJCC stage, N stage, and chemotherapy as independent prognostic risk factors for OS, while AJCC stage and N stage were found to be independent prognostic risk factors for CSS,

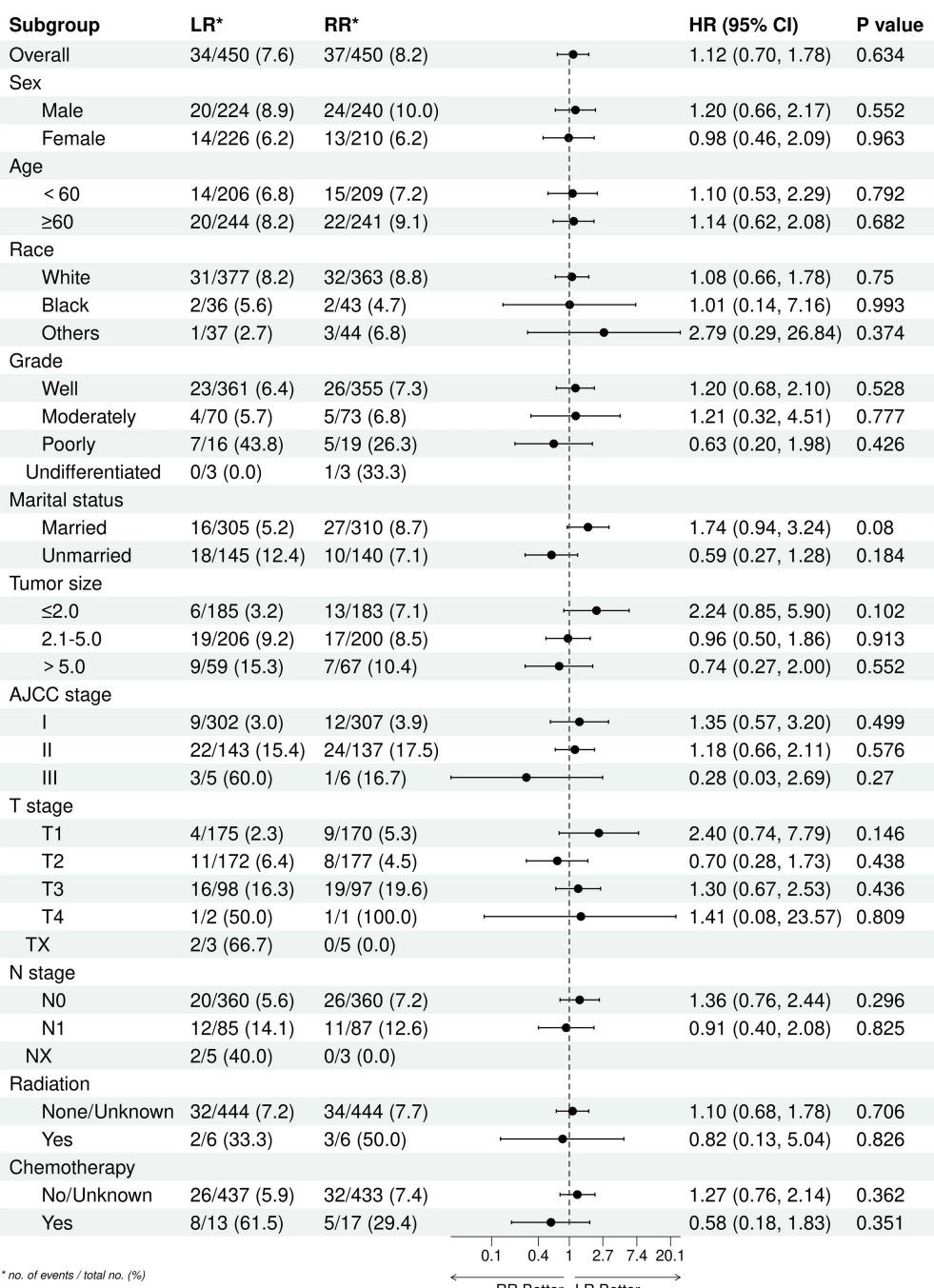

| Subgroup | LR* | RR* | HR (95% CI) | P value |
|---|---|---|---|---|
| Overall | 34/450 (7.6) | 37/450 (8.2) | 1.12 (0.70, 1.78) | 0.634 |
| Sex | | | | |
| Male | 20/224 (8.9) | 24/240 (10.0) | 1.20 (0.66, 2.17) | 0.552 |
| Female | 14/226 (6.2) | 13/210 (6.2) | 0.98 (0.46, 2.09) | 0.963 |
| Age | | | | |
| < 60 | 14/206 (6.8) | 15/209 (7.2) | 1.10 (0.53, 2.29) | 0.792 |
| ≥60 | 20/244 (8.2) | 22/241 (9.1) | 1.14 (0.62, 2.08) | 0.682 |
| Race | | | | |
| White | 31/377 (8.2) | 32/363 (8.8) | 1.08 (0.66, 1.78) | 0.75 |
| Black | 2/36 (5.6) | 2/43 (4.7) | 1.01 (0.14, 7.16) | 0.993 |
| Others | 1/37 (2.7) | 3/44 (6.8) | 2.79 (0.29, 26.84) | 0.374 |
| Grade | | | | |
| Well | 23/361 (6.4) | 26/355 (7.3) | 1.20 (0.68, 2.10) | 0.528 |
| Moderately | 4/70 (5.7) | 5/73 (6.8) | 1.21 (0.32, 4.51) | 0.777 |
| Poorly | 7/16 (43.8) | 5/19 (26.3) | 0.63 (0.20, 1.98) | 0.426 |
| Undifferentiated | 0/3 (0.0) | 1/3 (33.3) | | |
| Marital status | | | | |
| Married | 16/305 (5.2) | 27/310 (8.7) | 1.74 (0.94, 3.24) | 0.08 |
| Unmarried | 18/145 (12.4) | 10/140 (7.1) | 0.59 (0.27, 1.28) | 0.184 |
| Tumor size | | | | |
| ≤2.0 | 6/185 (3.2) | 13/183 (7.1) | 2.24 (0.85, 5.90) | 0.102 |
| 2.1-5.0 | 19/206 (9.2) | 17/200 (8.5) | 0.96 (0.50, 1.86) | 0.913 |
| > 5.0 | 9/59 (15.3) | 7/67 (10.4) | 0.74 (0.27, 2.00) | 0.552 |
| AJCC stage | | | | |
| I | 9/302 (3.0) | 12/307 (3.9) | 1.35 (0.57, 3.20) | 0.499 |
| II | 22/143 (15.4) | 24/137 (17.5) | 1.18 (0.66, 2.11) | 0.576 |
| III | 3/5 (60.0) | 1/6 (16.7) | 0.28 (0.03, 2.69) | 0.27 |
| T stage | | | | |
| T1 | 4/175 (2.3) | 9/170 (5.3) | 2.40 (0.74, 7.79) | 0.146 |
| T2 | 11/172 (6.4) | 8/177 (4.5) | 0.70 (0.28, 1.73) | 0.438 |
| T3 | 16/98 (16.3) | 19/97 (19.6) | 1.30 (0.67, 2.53) | 0.436 |
| T4 | 1/2 (50.0) | 1/1 (100.0) | 1.41 (0.08, 23.57) | 0.809 |
| TX | 2/3 (66.7) | 0/5 (0.0) | | |
| N stage | | | | |
| N0 | 20/360 (5.6) | 26/360 (7.2) | 1.36 (0.76, 2.44) | 0.296 |
| N1 | 12/85 (14.1) | 11/87 (12.6) | 0.91 (0.40, 2.08) | 0.825 |
| NX | 2/5 (40.0) | 0/3 (0.0) | | |
| Radiation | | | | |
| None/Unknown | 32/444 (7.2) | 34/444 (7.7) | 1.10 (0.68, 1.78) | 0.706 |
| Yes | 2/6 (33.3) | 3/6 (50.0) | 0.82 (0.13, 5.04) | 0.826 |
| Chemotherapy | | | | |
| No/Unknown | 26/437 (5.9) | 32/433 (7.4) | 1.27 (0.76, 2.14) | 0.362 |
| Yes | 8/13 (61.5) | 5/17 (29.4) | 0.58 (0.18, 1.83) | 0.351 |

*no. of events / total no. (%)*

0.1  0.4  1  2.7  7.4  20.1

← RR Better    LR Better →

**Fig 4. Subgroup analysis of CSS between LR group and RR group after PSM.**

but not the surgical method. This study suggests that LR may be a sufficient treatment option for patients with PNEC without distant metastasis.

It is generally advisable to opt for surgery in cases of functional PNETs, nonfunctional PNETs greater than 2.0 cm, or symptomatic nonfunctional PNETs [18]. PNEC can broadly be divided into three categories: resectable, locally advanced, and metastatic. According to a study by Crippa et al. [19], surgical removal is vital in predicting the prognosis of resectable

PNEC. Similarly, a multicenter retrospective study indicated that surgical intervention can positively impact the prognosis of patients diagnosed with high-grade PNEC. For certain patients presenting with both localized and metastatic high-grade PNEC, radical surgical intervention should be taken into account [20]. Apart from surgical treatment, our findings suggest that chemotherapy also holds significant importance in influencing the OS of patients with resectable PNEC. Individuals who have undergone surgical excision for PNEC are susceptible to experiencing local or distant recurrences. Hence, Sorbye et al. [21] proposed that post-surgery, adjuvant chemotherapy should be administered to enhance the prognosis of PNEC patients. The NANETS guidelines endorse a 4-6 cycle adjuvant therapy regime incorporating cisplatin or carboplatin alongside etoposide for patients who have undergone PNEC resection [13].

In our investigation, we solely incorporated individuals with PNEC who exhibited no signs of distant metastases, excluding those whose cancer had spread to distant organs like the liver, lungs, or bones. Several studies have indicated that removing the primary tumor surgically is advantageous for PNETs patients with distant metastases [22–25]. Nevertheless, PNETs are less virulent than PNEC, displaying lesser invasiveness and a reduced likelihood of metastasis. When total elimination of all metastases is unattainable, palliative surgery may still aid in alleviating symptoms, increasing longevity, and enhancing the overall wellbeing of patients. On the other hand, PNEC patients with distant metastases encounter a highly aggressive form of the disease, marked by significant invasiveness and a high metastatic potential, making it arduous to surgically remove all lesions entirely. Moreover, the surgical procedure can inflict considerable trauma, potentially leading to a challenging recovery process or even a worsening of the patient's condition. Hence, the utility of surgery in such instances remains a topic of debate. Consequently, for PNEC patients with distant metastases, numerous guidelines discourage surgical treatment and instead endorse chemotherapy as the preferred therapeutic approach [12–14].

Chen et al. [26] conducted a comparison of the outcomes for patients with 10mm to 20mm rectal NETs who underwent either LR or RR, and found no significant difference in OS and CSS between the two groups. Similarly, another study examined the prognosis of patients with rectal gastrointestinal stromal tumors treated with LR and RR, and also reported equivalent disease-free survival (DFS) and OS for both groups [27]. In our research, we assessed the prognosis of PNEC patients without distant metastases who received either LR or RR. In this context, LR mainly consisted of local tumor removal or partial pancreatic resection, whereas RR involved more extensive procedures such as partial pancreatic resection combined with duodenal resection, total pancreatic resection, or total pancreatic resection with partial gastric or duodenal resection. Our findings also indicated similar OS and CSS between the LR and RR groups. Prior research has identified lymph node metastasis as a key factor influencing the prognosis of PNENs patients [28, 29]. Consistent with these findings, our study also revealed that N stage was a significant predictor of OS and CSS in PNEC patients, with a poorer prognosis for those with lymph node metastasis ($P < 0.05$). Nevertheless, we observed that the LR and RR groups had comparable OS and CSS, irrespective of the presence of lymph node metastasis. We hypothesize that the similar prognosis between the LR and RR groups in PNEC patients without distant metastases may be due to the following reasons: LR primarily entails local tumor excision or partial pancreatic resection, which is less invasive, facilitates quicker postoperative recovery, and retains more normal pancreatic tissue and other organ functions. While RR involves a wider resection area to achieve radical removal, in the absence of distant metastases, this more extensive resection may not offer additional survival advantages. Moreover, RR is associated with greater surgical trauma and a higher risk of postoperative complications, which can negatively impact patients' quality of life. It's also worth noting that surgery

is often not the sole treatment for PNEC. The use of adjuvant therapies, such as postoperative chemotherapy and radiotherapy, is also vital for enhancing patient survival and potentially improving prognosis.

Our research encounters several constraints. Initially, our analysis is retrospective and depends on the SEER database, which inherently bears the risk of inconsistent and biased data. Secondly, the SEER database is deficient in thorough information pertaining to postoperative complications, specific chemotherapy protocols, disease recurrence, and the individualized criteria for surgical method selection. These factors could significantly affect patients' long-term outcomes. Thirdly, this study lacks detailed information on surgical techniques, surgical experience, and pre- and post-operative pancreatic function management, which may provide a comprehensive understanding of the surgical approaches. Lastly, it's essential to recognize that the SEER database mainly mirrors the American healthcare system, and medical practices can differ widely among various regions and cultures. Hence, prudence is required when generalizing our study's results to other geographic or cultural settings. However, to our understanding, this study represents the most extensive comparison of long-term outcomes between LR and RR treatments for PNEC conducted thus far. Additionally, we utilized PSM to mitigate potential biases and performed subgroup analyses to assess the impact of different factors on patient prognosis.

In conclusion, our research indicated that there was no advantage in survival rates with RR compared to LR for PNEC patients without distant metastasis, implying that LR might suffice as treatment for such individuals. This investigation serves as a foundation for clinicians to tailor surgical decisions for PNEC patients, ultimately optimizing patient outcomes.

## Supporting information

**Supplementary Table 1. Demographic and clinical characteristics of patients before multiple imputation.**
(DOCX)

## Author contributions

**Data curation:** Tingting Xia.

**Formal analysis:** Jinhao Li.

**Funding acquisition:** Zhengqing Liu, Xuehua Jiao, Zhenguo Qiao.

**Investigation:** Zhengqing Liu.

**Methodology:** Xuehua Jiao.

**Resources:** Chunfang Xu.

**Software:** Liangfu Han.

**Supervision:** Zhenguo Qiao.

**Writing – original draft:** Tingting Xia, Jinhao Li.

**Writing – review & editing:** Zhenguo Qiao, Chunfang Xu, Liangfu Han.

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
