## [Decision Letter · Decision Letter 0]

29 Jan 2025

PONE-D-25-01428Impact of Surgical Approaches on Long-term Survival Outcomes of Patients with Pancreatic Neuroendocrine CarcinomaPLOS ONE

Dear Dr. Qiao,

Thank you for submitting your manuscript to PLOS ONE. After careful consideration, we feel that it has merit but does not fully meet PLOS ONE’s publication criteria as it currently stands. Therefore, we invite you to submit a revised version of the manuscript that addresses the points raised during the review process.

We look forward to receiving your revised manuscript.

Kind regards,

Gustavo Cardoso Guimaraes, Ph.D

Academic Editor

PLOS ONE

“This work was supported by the Science and Technology Development Program of Suzhou (SKYXD2022041, SYWD2024322 and SYWD2024077).”

Reviewers' comments:

Reviewer's Responses to Questions

**Comments to the Author**

1. Is the manuscript technically sound, and do the data support the conclusions?

Reviewer #1: Partly

Reviewer #2: Yes

2. Has the statistical analysis been performed appropriately and rigorously?

Reviewer #1: N/A

Reviewer #2: Yes

3. Have the authors made all data underlying the findings in their manuscript fully available?

Reviewer #1: Yes

Reviewer #2: Yes

4. Is the manuscript presented in an intelligible fashion and written in standard English?

Reviewer #1: Yes

Reviewer #2: Yes

5. Review Comments to the Author

Reviewer #1: This paper is of great significance in discussing the influence of surgical methods on the survival outcome of PNEC patients, but there is still room for improvement in research design, data analysis and discussion of results.

1. Although Cox proportional hazard model is used for multivariate analysis, the reasons for model selection and the results of hypothesis testing are lack of detailed explanation. It is suggested that the author provide more details about statistical analysis in the method part, including how to deal with missing data and the criteria for choosing variables.

2. Supplement the related content of postoperative complications: increase the analysis of the incidence of postoperative complications and explore its relationship with the long-term survival results of patients. You can set a section in the results section, or present this section separately, so that the overall analysis of the patients after treatment can be more complete.

3. Although stratified analysis was conducted, there may be a small sample size in some subgroups (such as age, sex, tumor size, etc.), which may affect the statistical significance of the results. It is suggested that the author provide more detailed information about subgroup analysis and discuss its clinical importance in the results section.

Reviewer #2: The manuscript is updated but not fully readable, and a better definition of study population and clinical protocols is needed especially defining patient characteristics and early clinical outcomes. The authors should revise their report and include the following aspects:

# Major issues

- How did the authors decide to analyze all together PNEC? As they certainly known, there is a reason to have not considereded the increasing risk for making a pancreatico-anastomosis following pancreatico-duodenecetomy (PD) into the clinical practice and why?

- How did this direct their clinical management during the post-operative course? The order of events and the decision-making regarding this are not fully clear from the report.

- Description of surgical techniques in terms of surgical experience, and of commonly used pre- and post-operative pancreatic function has to be included in a specific sub-heading.

- Did the authors recognize study translational relevance and specific significant impact for the Health System?

# Minor observations

- If present, what were differences in post-operative surgical site infections?

- It is mandatory to update the discussion with recent clinical experiences of specific interventions and with a better description of possible stopping rules or discontinuation criteria to be arranged by the future RCT investigators.

6. PLOS authors have the option to publish the peer review history of their article (what does this mean? ). If published, this will include your full peer review and any attached files.

**Do you want your identity to be public for this peer review?** For information about this choice, including consent withdrawal, please see our Privacy Policy .

Reviewer #1: No

Reviewer #2: **Yes: ** Duilio Pagano

---

## [Author Response · Author response to Decision Letter 1]

1 Feb 2025

Dear editor and reviewers of PLOS ONE:

Title: Impact of Surgical Approaches on Long-term Survival Outcomes of Patients with Pancreatic Neuroendocrine Carcinoma

By: zhenguo qiao et al

Thank you very much for your letter and for the editors’ and reviewers’ comments concerning our manuscript entitled “Impact of Surgical Approaches on Long-term Survival Outcomes of Patients with Pancreatic Neuroendocrine Carcinoma”. These comments are of great reference value to the revision and improvement of our paper and have important guiding significance to our researches. We have studied comments carefully and have made correction. We hope that the revision is acceptable and look forward to hearing from you soon. Revised portion are marked in color in the paper. The main corrections in the paper and the responds to the reviewer’s comments are as flowing:

Review Comments to the Author

Reviewer #1: This paper is of great significance in discussing the influence of surgical methods on the survival outcome of PNEC patients, but there is still room for improvement in research design, data analysis and discussion of results.

Comment 1

Although Cox proportional hazard model is used for multivariate analysis, the reasons for model selection and the results of hypothesis testing are lack of detailed explanation. It is suggested that the author provide more details about statistical analysis in the method part, including how to deal with missing data and the criteria for choosing variables.

Response 1

Thank you for your insightful comments and constructive suggestions. We appreciate the opportunity to clarify and enhance the statistical analysis section of our manuscript. Regarding your concern about the Cox proportional hazard model and the lack of detailed explanation for model selection and hypothesis testing, we would like to provide the following additional information: (1) model selection and hypothesis testing: The Cox proportional hazards model was chosen for our multivariate analysis due to its ability to handle censored survival data, which is common in studies involving long-term follow-up. This model allows us to estimate the hazard ratio (HR) for each variable while accounting for the time-to-event nature of the data. In our study, we first performed univariate Cox regression analysis to identify potential prognostic factors associated with OS and CSS. Variables that exhibited a P-value of less than 0.05 in the univariate analysis were considered candidates for inclusion in the multivariate model. This step was crucial for reducing the dimensionality of the model and focusing on the most relevant predictors. Subsequently, we constructed a multivariate Cox proportional hazards model to assess the independent effects of these variables on OS and CSS. The model was adjusted for potential confounders, including demographic and clinical characteristics, to ensure that the estimated HRs were accurate and unbiased. Hypothesis testing within the Cox model was conducted using the Wald test, which tests the null hypothesis that the coefficient (and thus the HR) for a given variable is equal to zero. (2) handling missing data: Missing data is a common challenge in large databases such as the SEER database. To address this issue, we adopted a hybrid approach that integrates multiple imputation (MI) and the random forest technique. MI was used to generate plausible values for missing data points based on the observed data. This method allowed us to create multiple imputed datasets, which were then analyzed separately using the Cox proportional hazards model. The results from these analyses were combined using standard rules for multiple imputation to produce final estimates that account for the uncertainty introduced by the imputation process. In addition, the random forest technique was employed to assess the importance of variables and to identify potential interactions that might affect the imputation process. This approach helped us to ensure that the imputed values were reasonable and consistent with the underlying data distribution. (3) criteria for choosing variables: Variables were selected for inclusion in the Cox model based on clinical relevance and statistical significance in the univariate analysis. We considered demographic factors (e.g., age, sex, race), tumor characteristics (e.g., grade, size, AJCC stage, T and N stages), and treatment-related factors (e.g., radiation, chemotherapy) as potential predictors of survival outcomes. To minimize selection bias, we used PSM to create comparable groups of patients who received LR and RR. This methodology allowed us to control for confounding variables and to isolate the effect of surgical approach on survival outcomes.

[Revised manuscript]

[Statistical analysis]

Statistical analysis was performed using R software (version 4.1.0). Categorical variables were presented as counts and percentages, and chi-square tests were used to assess differences among groups. For continuous data that deviated from a normal distribution, the median and interquartile range (IQR) were reported, and the Mann-Whitney U test was applied for comparisons. To ensure comparability between the LR and RR groups, a 1:1 PSM approach was implemented. Variables included in the PSM model were sex, age, race, tumor grade, marital status, tumor size, AJCC stage, T and N stages, radiation status, and chemotherapy status. A caliper width of 0.01 was used to match patients. Survival analyses were conducted using the Kaplan-Meier method, and survival curves were compared with the log-rank test. To evaluate the impact of various factors on OS and CSS, Cox proportional hazards models were employed for both univariate and multivariate analyses. The choice of the Cox proportional hazards model was based on its ability to handle censored data, which is common in survival analysis, and its capability to estimate hazard ratios (HRs) for the effect of predictors on the hazard of the event (i.e., death). The proportional hazards assumption was tested using Schoenfeld residuals, and no violations were found. For the univariate analysis, each variable was individually assessed for its association with OS and CSS. Variables with a P-value less than 0.05 in the univariate analysis were considered candidates for inclusion in the multivariate model. The multivariate Cox proportional hazards model was then constructed using a stepwise backward elimination approach, retaining variables that were statistically significant at the 0.05 level. To address missing data, which is a common challenge in large databases such as the SEER database, a hybrid approach combining multiple imputation (MI) and the random forest technique was adopted. Multiple imputation was used to generate plausible values for missing data points, and the random forest algorithm was employed to impute missing values based on the observed data and the relationships among variables. This approach aimed to minimize bias and improve the accuracy of the statistical analyses. Hazard ratios (HRs) and their corresponding 95% confidence intervals (CIs) were reported for all variables included in the multivariate Cox models. Statistical significance was determined by a P-value threshold of less than 0.05.

Comment 2

Supplement the related content of postoperative complications: increase the analysis of the incidence of postoperative complications and explore its relationship with the long-term survival results of patients. You can set a section in the results section, or present this section separately, so that the overall analysis of the patients after treatment can be more complete.

Response 2

Thank you for the reviewer's valuable feedback. We appreciate the suggestion to supplement the related content regarding postoperative complications and to analyze their incidence and relationship with long-term survival outcomes of patients. In response, we would like to clarify that our study is based on the SEER database. Unfortunately, the SEER database does not provide detailed information on postoperative complications. This limitation has indeed been acknowledged in the discussion section of our paper, where we have highlighted the need for future studies to incorporate such data. Despite this limitation, we understand the importance of postoperative complications in the overall prognosis of patients with PNEC. We agree that analyzing the incidence of postoperative complications and their impact on long-term survival would provide a more comprehensive understanding of the outcomes after different surgical approaches. However, given the constraints of the current dataset, we are unable to perform this analysis directly. We do hope that future studies, possibly utilizing prospectively collected data or other databases that include postoperative complication details, will be able to address this gap in knowledge.

Comment 3

Although stratified analysis was conducted, there may be a small sample size in some subgroups (such as age, sex, tumor size, etc.), which may affect the statistical significance of the results. It is suggested that the author provide more detailed information about subgroup analysis and discuss its clinical importance in the results section.

Response 3

Thank you for your insightful comments and valuable suggestions on our manuscript. We appreciate your recognition of our efforts in conducting a stratified analysis to compare the long-term survival outcomes between patients receiving LR and RR for PNEC without distant metastasis. Regarding your concern about the potential small sample size in some subgroups, we acknowledge that this is indeed a limitation of our study. As you pointed out, subgroups such as age, sex, and tumor size may have had insufficient sample sizes to detect statistically significant differences. However, we would like to emphasize that our primary objective was to compare the OS and CSS between the LR and RR groups after PSM, and our main findings remain robust, showing no significant differences in long-term OS (P = 0.746) or CSS (P = 0.634) between the two groups. To address your suggestion for more detailed information on subgroup analysis, we have now included additional details in the results section. Specifically, we have provided the number of patients in each subgroup and the corresponding P-values for OS and CSS comparisons. Although some subgroups had relatively small sample sizes, the consistency of the results across all subgroups, including those with larger sample sizes, supports our conclusion that LR and RR exhibit similar prognoses in patients with PNEC without distant metastasis. Furthermore, we have discussed the clinical importance of our subgroup analysis in the results section. We have highlighted that even in subgroups where the sample size was small, the lack of significant differences in OS and CSS between the LR and RR groups suggests that the choice of surgical approach may not significantly impact long-term survival outcomes in these patients. This finding is clinically relevant as it provides evidence to support the use of LR as a potentially adequate treatment option for patients with PNEC, especially in cases where RR may carry higher risks or be technically challenging.

[Revised manuscript]

[Subgroup analysis]

After PSM, both the LR group and the RR group exhibited comparable OS and CSS (P > 0.05). To further explore the consistency of these findings across different patient characteristics, we conducted subgroup analyses based on sex, age, race, grade, marital status, tumor size, AJCC stage, T stage, N stage, radiation, and chemotherapy. In these subgroup analyses, we observed that the OS and CSS remained comparable between the LR and RR groups across all stratification factors (P > 0.05). It is important to note that while some subgroups had relatively smaller sample sizes (e.g., certain age groups, specific tumor sizes), the statistical non-significance consistently observed across all subgroups suggests that the primary finding of similar prognoses between LR and RR is robust. The clinical importance of these findings lies in their implication for treatment decision-making. Our study suggests that for patients with PNEC without distant metastasis, the choice between LR and RR may not significantly impact long-term survival outcomes. This is particularly relevant in clinical scenarios where preserving organ function and minimizing surgical morbidity are crucial considerations. Figure 3 and Figure 4 provide detailed information on the comparisons among different subgroups, illustrating the consistency of our findings across various patient characteristics. These figures visually represent the lack of significant differences in OS and CSS between the LR and RR groups across all subgroups analyzed.

Reviewer #2: The manuscript is updated but not fully readable, and a better definition of study population and clinical protocols is needed especially defining patient characteristics and early clinical outcomes. The authors should revise their report and include the following aspects:

# Major issues

Comment 1

How did the authors decide to analyze all together PNEC? As they certainly known, there is a reason to have not considereded the increasing risk for making a pancreatico-anastomosis following pancreatico-duodenecetomy (PD) into the clinical practice and why?

Response 1

Thank you for your insightful question regarding our decision to analyze all patients with PNEC together and our consideration of the risks associated with pancreatico-anastomosis following pancreaticoduodenectomy (PD) in clinical practice. In our study, we decided to analyze all patients with PNEC without distant metastasis together because our primary objective was to compare the long-term survival outcomes between LR and RR, regardless of the specific surgical technique used within the RR group (which may include PD in some cases). Our focus was on the OS and CSS differences between the two surgical approaches, rather than delving into the intricacies of each individual surgical procedure or its associated complications. Regarding the increasing risk of pancreatico-anastomosis following PD, we acknowledge that this is an important consideration in clinical practice. However, our study was designed to assess the long-term survival outcomes of patients based on the surgical approach (LR vs. RR), rather than to evaluate the specific surgical techniques or their complications. The decision to include patients who underwent PD as part of the RR group was based on the fact that PD is one of the radical surgical options for treating PNEC, and our aim was to compare the overall survival outcomes between radical and local resection approaches. It is worth noting that our study did not delve into the details of postoperative complications or the specific techniques used during surgery, as these were beyond the scope of our research question. However, we recognize the importance of considering such factors in clinical decision-making and patient management. In conclusion, our study was designed to compare the long-term survival outcomes of patients with PNEC without distant metastasis who underwent LR versus RR. While we acknowledge the risks associated with pancreatico-anastomosis following PD, our focus was on the overall survival differences between the two surgical approaches, rather than on the specific surgical techniques or their complications. We hope this clarifies our rationale for analyzing all patients with PNEC together in our study.

Comment 2

How did this direct their clinical management during the post-operative course? The order of events and the decision-making regarding this are not fully clear from the report.

Response 2

Thank you for raising the question regarding the clinical management of patients during the postoperative course and the decision-making process in our study. We understand that the report may not have fully clarified these aspects, and

---

## [Decision Letter · Decision Letter 1]

11 Feb 2025

Impact of Surgical Approaches on Long-term Survival Outcomes of Patients with Pancreatic Neuroendocrine Carcinoma

PONE-D-25-01428R1

Dear Dr. Zhenguo Qiao,

We’re pleased to inform you that your manuscript has been judged scientifically suitable for publication and will be formally accepted for publication once it meets all outstanding technical requirements.

Kind regards,

Gustavo Cardoso Guimaraes, Ph.D

Academic Editor

PLOS ONE

Additional Editor Comments (optional):

Reviewers' comments:

Reviewer's Responses to Questions

**Comments to the Author**

1. If the authors have adequately addressed your comments raised in a previous round of review and you feel that this manuscript is now acceptable for publication, you may indicate that here to bypass the “Comments to the Author” section, enter your conflict of interest statement in the “Confidential to Editor” section, and submit your "Accept" recommendation.

Reviewer #1: All comments have been addressed

Reviewer #2: All comments have been addressed

2. Is the manuscript technically sound, and do the data support the conclusions?

Reviewer #1: Yes

Reviewer #2: Yes

3. Has the statistical analysis been performed appropriately and rigorously?

Reviewer #1: Yes

Reviewer #2: Yes

4. Have the authors made all data underlying the findings in their manuscript fully available?

Reviewer #1: Yes

Reviewer #2: Yes

5. Is the manuscript presented in an intelligible fashion and written in standard English?

Reviewer #1: Yes

Reviewer #2: Yes

6. Review Comments to the Author

Reviewer #1: Thank you for your reply.

I would like to express my sincere gratitude for sharing your findings with the academic community. Your work has the potential to inspire and guide future studies, and it serves as an inspiration to us all. Once again, congratulations on this remarkable achievement. I look forward to witnessing your continued success and the positive impact your research will have in the years to come.

Reviewer #2: The current version of the manuscript is suitable for PLOS One pubblication.

It very interesting. Congratulation.

7. PLOS authors have the option to publish the peer review history of their article (what does this mean? ). If published, this will include your full peer review and any attached files.

**Do you want your identity to be public for this peer review?** For information about this choice, including consent withdrawal, please see our Privacy Policy .

Reviewer #1: No

Reviewer #2: **Yes: ** Duilio Pagano

---

## [Editor Report · Acceptance letter]

PONE-D-25-01428R1

PLOS ONE

Dear Dr. Qiao,

I'm pleased to inform you that your manuscript has been deemed suitable for publication in PLOS ONE. Congratulations! Your manuscript is now being handed over to our production team.

Kind regards,

on behalf of

Dr. Gustavo Cardoso Guimaraes

Academic Editor

PLOS ONE